# Probiotic Properties of Lactic Acid Bacteria Newly Isolated from Algerian Raw Cow’s Milk

**DOI:** 10.3390/microorganisms11082091

**Published:** 2023-08-15

**Authors:** Nacima Kouadri Boudjelthia, Meryem Belabbas, Nahla Bekenniche, Magali Monnoye, Philippe Gérard, Ali Riazi

**Affiliations:** 1Laboratory of Beneficial Microorganisms, Functional Food and Health, Department of Food Sciences, Faculty of Natural and Life Sciences, Abdelhamid Idn Badis University, Mostaganem 27000, Algeria; nacima.kouadriboudjelthia@univ-mosta.dz; 2Department of Agronomy Sciences, Faculty of Natural and Life Sciences, DjillaliLiabès University, Sidi Bel Abbes 22000, Algeria; meryem.belabbas@univ-sba.dz; 3Laboratory of Food Technology and Nutrition, Department of Biology, Faculty of Natural and Life Sciences, Abdelhamid Ibn Badis University, Mostaganem 27000, Algeria; nahla.bekenniche@univ-mosta.dz; 4Micalis Institute, INRAE, AgroParisTech, Paris-Saclay University, 78350 Jouy-en-Josas, France; magali.monnoye@inrae.fr (M.M.); philippe.gerard@inrae.fr (P.G.)

**Keywords:** lactic acid bacteria (LAB), raw cow’s milk, probiotic, antioxidant, antibacterial

## Abstract

This study aims to screen new LAB from Algerian cow’s milk to assess their probiotic properties. Molecular identification and MALDI-TOF mass spectrometry methods were used to identify the LAB isolates. The probiotic potential of isolates was determined with in vitro tests of survival to gastrointestinal conditions (pH 2, 0.3% pepsin, 0.5% bile salts, 0.1% trypsin, and 0.1% pancreatic amylase) and antimicrobial and antioxidant activities. Eight isolates were identified as *Lactiplantibacillus plantarum* (100%) and one isolate as *Lacticaseibacillus rhamnosus* (95.75%). The MALDI-TOF MS analysis of the isolates confirms that the strains belong to the group of lactobacilli bacteria, particularly *Lactiplantibacillus plantarum*. The high survival rate reflects a good strain tolerance to the in vitro host simulated gastrointestinal conditions. All bacteria exhibit an antibacterial activity strain with inhibition zone diameters ranging from 4.9 mm against *Aspergillus niger* ATCC 106404 to 17.47 mm against *Candida albicans* ATCC 10231. The antioxidant activity with the highest DPPH scavenging activity (92.15%) was obtained with the LbN09 strain. In light of these results, some of the strains isolated from raw milk of the local Algerian breed cows show promising probiotic properties, giving them a possible use in preserving food from microbial spoilage and oxidation during storage.

## 1. Introduction

The interest in probiotics is illustrated by the constant increase in the world market for these living microorganisms, which are increasingly sought after by consumers who are more aware of their well-being. The relevance of probiotics in maintaining and improving human health is related to their beneficial effect on intestinal flora. The screening of new bacterial strains with probiotic properties is costly and time-consuming. Raw cow’s milk is a potential source of LAB, which is one of the most commonly used bacteria that can provide food and health benefits related to their ability to produce secondary metabolites and functional proteins [1]. Many LABs are considered probiotics and are targeted by several researchers for their immense genetic, metabolic, and ecological diversity, as they colonize a wide range of habitats and can have different industrial applications [2].

Probiotics are defined as live bacteria if administrated in adequate amounts and provide health benefits [3]. These probiotics are isolated from several ecological niches, in particular, from cow’s milk, which is an important source of LAB, in particular, species belonging to the *Lactobacillus* genus. Probiotic strains prevent intestinal infections and dysbiosis in the host by producing lactic acid and functional proteins (i.e., bacteriocins) able to exert an antagonistic activity against the pathogenic bacteria; they also contribute to the maintenance of the essential balance of intestinal microbiota [4,5].

Probiotic bacteria cannot exert their beneficial effects if they cannot survive digestive hostilities (barriers of gastric acidity, bile salts, and digestive enzymes) during their transit through the gastrointestinal tract. Their adherence to the intestinal cells can also contribute to their probiotic properties [6]. Several studies have shown beneficial effects in the host of most *Lactobacillus plantarum* species isolated from milk and fermented food, and which have been evaluated for their probiotic potential through their resistance to digestive hostilities, antioxidant and antagonistic activities, and adherence to epithelial cells [7,8].

Moreover, several specific new LAB strains were selected for the prevention and/or in vivo treatment of several inflammatory bowel diseases due to food poisoning [9,10]. The initial classification of *lactobacilli* was based on morphological and physiological characteristics (gram, cell morphology, colony appearance); on growth parameters (optimum incubation temperature, pH tolerance, and oxygen requirements), on biochemical characterization, the used substrates, and the final metabolic products made it possible to divide the *lactobacilli* into three groups according to their fermentative pathway: obligate homofermentative, facultative homofermentative, and obligate heterofermentative [11]. Obligate homofermentative lactobacilli use the Embden-Meyerhof-Parnas (EMP) pathway to convert glucose into lactic acid but cannot use the pentose-phosphate pathway to ferment pentoses. Obligate heterofermentative lactobacilli exclusively use the pentose-phosphate pathway; while facultative homofermentative use EMP but also ferment pentoses [11].

Molecular taxonomy, based on DNA sequencing, allows sequences to be compared to a database for species identification, and the phylogenetic tree represents the relatedness of lactobacilli strains according to divergence from common ancestors [12]. The MALDI-TOF-MS identification method is currently applied to the routine identification of many microorganisms such as the reliable and rapid identification of LAB [13]. In this study, nine LAB isolates from Algerian raw cow’s milk were assessed. Isolates were first phenotypically selected and identified by 16S rDNA and MALDI-TOF MS before evaluating their probiotic properties in vitro, including antioxidant and antimicrobial activities as well as their ability to survive simulated gastrointestinal conditions (gastric fluid, bile salts, trypsin, and pancreatic amylase). Algerian cow’s milk is lacking information as a potential medium LAB that may be a source of probiotic strain isolation, so this study is the first attempt to screen such new LAB strains with probiotic status.

## 2. Materials and Methods

### 2.1. Sample Collection

Ten milk samples were collected from ten cows in aseptic conditions with manual milking using sterile gloves after cleaning the udders and surrounding skin with sterile water, throwing the first drops, and then transporting them in an ice box to the laboratory for immediate isolation procedure. Cows were living in a Menasria traditional livestock farm, district of Mascara located in Western Algeria (36.17547° N, 1.44178° W), and fed with fodder, bran, and natural grasses from pasture.

### 2.2. Strains, Growth Conditions, and Identification

Milk samples were enriched in skimmed milk (10%) and then diluted and plated on MRS Agar (Conda laboratorios, Madrid, Spain) [14]. Next, 10 mL of enriched milk was added aseptically to 90 mL of sterile saline solution (0.9%). Serial dilutions were made using sterile saline solution in sterile tubes up to 10^−4^. Sterile MRS plate agar was inoculated (1 mL) and incubated anaerobically in a CO_2_ incubator at 37 °C for 24 h. Bacterial purification was routinely assessed using morphological tests such as gram staining, catalase, CO_2_ production, and capacity of growth in different NaCl concentrations for a preliminary selection of different isolates of lactobacilli. Gram-positive, catalase-negative bacilli isolates selected were presumed to be lactobacilli and were stored in MRS-Glycerol (25/75 *v*/*v*) at –80 °C.

Genomic DNA of bacterial strains was extracted using the commercial extraction kit Promega Wizard^®^ Genomic DNA Purification Kit (Promega; Madison, WI, USA) according to the manufacturer’s recommendation. The cells were harvested from fresh bacterial pellets after culture centrifugation and supernatant removal. The quantity and the quality of DNA extracts were monitored using a NanoDrop spectrophotometer (Thermos Scientific, Wilmington, DE, USA). A Bio-Rad T100 PCR programmable Thermal Cycler was used to amplify the 16Sr RNA gene with prokaryotic 16S rRNA universal primer’s region V6-V8:P3:Bact968-GC-f (5′CGCCCGGGGCGCGCCCCGGGCGGGGCGGGGGCACGGGGGGAACGCGAAGAACCTTAC3′) and P4: Bact1401-r (3′GCGTGTGTACAAGACCC5′) as described by Nübelet al. [15]. The enzyme used was a Taq polymerase from Qbiogen (Qbiogen, Irvine, CA, USA) according to the manufacturer’s recommendations. The PCR mixes contained 10 to 50 ng/µL DNA matrix, 1 µL of each primer, 3 µL of MgCl_2_ solution, 25 µL of TaqMix (Taq polymerase + dNTPs), and H_2_O 50 µL. PCR amplification protocol was carried out with an initial denaturation at 95 °C for 15 min, followed by 30 denaturing cycles at 97 °C for 1 min, annealing at 58 °C for 1 min and extension at 72 °C for 1 min and 30 s. The procedure was completed with a final elongation step at 72 °C for 15 min.

16S rRNA amplified samples were purified by using a Wizard^®^ SV Gel and PCRClean-Up System Kit (Qiagen, Hilden, Germany) according to the manufacturer’s instructions and then hot start sequenced by a German sequencing company (Eurofins Genomic sequencing, Cologne, Germany). All obtained Fasta sequences were blasted to the BLAST database program of NCBI (http://blast.ncbi.nlm.nih.gov/Blast.cgi, accessed on 3 March 2022), and compared with the sequences deposited in GenBank for strain identification. A phylogenetic tree based on the analysis of 16S rRNA gene sequences was constructed using MEGA-X: Molecular Evolution Genetic Analysis [16] available at http://www.megasoftware.net. The distances were calculated using the Neighbor-joining method with jukes-cantor correction [17].

### 2.3. MALDI-TOF MS Identification

A single colony grown on MRS agar was transferred onto a MALDI-TOF steel target plate (MS Big Ankor 24BC, Bruker daltonics inc., Bremen, Germany). Next, 10 µL of a saturated solution of trans-3,5-dimethyloxy-4-hydroxycinnamic acid (HCCA portioned, Bruker) were added to 0.1µL of formic acid to allow crystallization of the matrix with the sample [13]. The plate was left to dry at ambient temperature and introduced into the mass spectrophotometer (Matrix Assisted Laser Desorption/Ionization-Time-Of Flight Mass Spectrometry: MALDI-TOF/MS; Microflex, Bruker). The automated system generates spectra interpreted by software (MALDI Biotyper TM4.0 Data base Brucker Daltonics, Bremen, Germany). The results are shown as the top 10 identification matches along with confidence scores ranging from 0.00 to 3.00. Data are representative of two experiments.

### 2.4. Strain Survival to the In Vitro Simulated Gastrointestinal Conditions

Survival capacity at the stomach transit was determined by Vizoso Pinto et al. [18] method. The bacteria were subjected to in vitro simulated gastric fluid and whose composition (%: P/V)) is: 6.2 NaCl, 0.22 CaCl_2_, 1.2 NaHCO_3_, 2.2 KCl, 0.3 pepsin, snd pH 2.5. The simulated gastric fluid was sterilized by filtering through a 0.22 µm membrane. Viable cells were counted with the agar plate method on MRS at the end of the time required for the process of converting the bolus into chyme in the stomach, which is 4 h of incubation at 37 °C [19]. The survival rate was calculated by the following formula:Survival rate (%) = (log UFC at time t_1_/log UFC at time t_2_) × 100 (1)

Survival strains of bile salts were assessed according to the method of Hyronimus et al. [20]. Briefly, 1 mL of each overnight culture (18 h) was centrifuged at 8000 rpm/10 min and the supernatant was discarded. Next, 50 µL of each bacterial pellet was inoculated in 450 µL of MRS (pH8 + bile salts (0.5%). The survival of lactobacilli strains was assessed by counting plate agar media at 4 h at 37 °C. This incubation was chosen based on the time required for digestion in the small intestine with bile salts [19]. The survival rate was calculated using the previous Formula (1).

Strain survival rate for digestive enzymes was assessed as described by Ouwehand et al. [21]. To do so, 50 µL of each strain suspension were inoculated into 450 µL of pancreatic amylase (Sigma, Aldrich, Neustadt, Germany) solution freshly prepared (0.1 mg/mL in PBS, pH 6.9) on one hand, and into 450 µL of trypsin (Alfa Aesar, by Thermo Fisher Scientific, Wilmington, DE, USA) solution freshly prepared (0.1 mg/mL in PBS, pH 7.6). Viable cells were determined with the MRS agar plate count method after 5 h exposure to enzymes at 37 °C. The survival rate was calculated using the previous Formula (1).

### 2.5. Determination of Biological Activities of the Strains

#### 2.5.1. Antimicrobial Activity

Antimicrobial activity of LABs strains against some relevant human pathogens including *Escherichia coli* ATCC 25922, *Staphylococcus aureus* ATCC 33862, *Pseudomonas aeruginosa* ATCC 27853, *Bacillus subtilis* ATCC 6633, *Candida albicans* ATCC 10231, and *Aspergillus niger* ATCC 106404 was determined as described by Chaalel et al. [22]. Overnight cultures strains in MRS broth medium at 37 °C were centrifuged for 10 min at 14,000× *g*. Next, 50 µL of filtered (0.2 µm filter) and neutralized supernatant were put in 5 mm diameter wells on MRS agar plate pre-inoculated with the indicators of pathogen bacteria. The agar plates were incubated at 37 °C overnight for all pathogenic bacteria tested, and for 48 to 72 h for the fungi *Candida albicans* ATCC 10231 and *Aspergillus niger* ATCC 106404. The clear zone formation indicates the positive antimicrobial activity of the supernatant metabolites against the pathogens [22].

#### 2.5.2. Antioxidant Activity

DPPH radical scavenging activity of LAB strains was determined according to the original method of Blois et al. [23]. Utilizing this method, 1 mL of each 18 h bacterial young culture was mixed with 1 mL of 0.004% (P/V) DPPH methanolic solution and left for 30 min in the dark at room temperature. Thereafter, each mixture was centrifuged at 3000 rpm for 10 min and the absorbance of the supernatant was measured at 517 nm against distilled water (negative control) in comparison to ascorbic acid (Sigma, Aldrich) as standard (positive control). The percentage inhibition (I%) of DPPH radical scavenging activity was calculated according to the following formula:I (%) = [(A_control_ − A_assay_)/A_control_)] × 100(2)

### 2.6. Statistical Analysis

The data obtained were analyzed using IBM SPSS statistics, version 26.0. Data were reported as mean ± SD. The level of statistical significance was set at *p* < 0.05 for the two-sided tests. The unidirectional ANOVA followed by the Tukey test was used to compare the results of survival of the different strains to the in vitro simulated gastrointestinal conditions and the antioxidant activity. The multidirectional MANOVA, followed by the Tukey test, was used to compare the results of the antimicrobial activity of the different strains obtained in the strain growth zone inhibition test with those of the reference antimicrobials.

## 3. Results

### 3.1. Analysis of a Total of Nine Selected LAB Isolates Reveals Eight Putative Strains Lactiplantibacillus plantarum and One Putative Strain Lacticaseibacillus rhamnosus, as Confirmed by Molecular (PCR-16) and Proteomic (MALDI-TOF) Identification

Phenotypic characterization of a total of nine obtained isolates revealed that all the cow’s milk isolated strains were gram-positive and catalase negative. However, to assign reliable profiles to the selected isolates, molecular identification remains essential and obligatory.

The 16S rRNA gene sequences of strains LbN1, LbN5, LbN9, LbN10, LbN11, LbN12, LbN13, LbN14, and LbN15 were submitted to NCBI Genbank database under the following number: OM976615, OM976616, OM976617, ON008307, ON008308, ON008309, ON008310, OM976618, and ON008311, respectively. All strains selected by PCR16s revealed the presence of bands from the DNA amplification as shown by the electrophoresis gel profile in Figure 1. The expected Amplicon length is about 434 bp fragment of the hypervariable V6-V8 region [24]. The crude sequences obtained from Eurofins Genomic sequencing were blasted using the BLAST database program of NCBI (http://blast.ncbi.nlm.nih.gov/Blast.cgi, accessed on 3 March 2022). The FASTA sequences were compared with the sequences deposited in GenBank for the identification of strains. Similarity scores (%) are presented in Table 1. The BLAST of Crude Sequences of the rRNA16S gene obtained by the universal P3:Bact968-GC-f sense primers and antisense primer P4: Bact1401-r, allowed the selection of 9 isolates belonging to the 2 genera *Lactiplantibacillus* sp. and *Lacticaseibacillus* sp. with 100 and 99.75% similarity, respectively, according to the Genbank data.

The Dendrogram reported in Figure 2 results from the alignment of the 16s rRNA sequences of the nine lactobacilli isolates and the sequences of the type strains retrieved from the NCBI database (*L. plantarum*, *L. argentoratensis*, *L. paraplantarum*, *L. casei*, *L. rhamnosus*, *L. pentosus*, *L. paracaseisubsptolerans*, *L. herbarum*). Analysis of phylogenetic distances grouped together eight isolates (LbN5, LbN9, LbN10, LbN11, LbN12, LbN13, LbN14, LbN15) in *Lactiplantibacillus plantarum* and one isolate (LbN1) in *Lacticaseibacillus rhamnosus*. The LbN1 strain has been grouped around the following standard strains: *Lacticaseibacillus rhamnosus* NBRC 3425, *Lacticaseibacilluscasei* ATCC 393, *Lacticaseibacillusparacasei* subsp. paracasei JCM 8130, and *Lacticaseibacillusparacaseisubsptolerans* NBRC 15906. The LbN12 strain has been identified in the group of the following standard strains: *Lactiplantibacillusherbarum* TCFE 032-E4, *Lactiplantibacillusparapalantarum* DSM 10667, *Lactiplantibacillus plantarum* Subsp. argentoratensis DK022, and *Lactiplantibacilluspentosus* ATCC 8041.

MALDI-TOF MS analysis concerned eight isolates (LbN5, LbN9, LbN10, LbN11, LbN12, LbN13, LbN14, and LbN15). The use of the MALDI-TOF/MS mass spectrometry method was necessary to support the molecular identity of isolates for which 16S rRNA identification was inconclusive. The identification list proposed by MALDI BioTyper 3.0 software (Bruker Daltonics) identified the strains as L. plantarum with the following scores values of identification: 2.302, 2.191, 2.050, 2.067, 2.205, 2.307 for strains LbN9, LbN10, LbN11, LbN12, LbN13, and LbN14, respectively; while scores values for the LbN5 and LbN15 strains were 1.924 and 1.868, respectively. This result indicates good reliability and a high probability of identification according to the criteria proposed by the constructor: a log (score) below 1.69 did not allow reliable identification; a log (score) between 1.70 and 1.99, allowed a probable identification of the genus with a low level of confidence; a log(score) between 2.00 and 2.29 indicates secure gender identification; and a log (score) between 2.30 and 3 indicates high probable species identification with high confidence.

### 3.2. The Isolated Strains Have Shown an Interesting Probiotic Profile

#### 3.2.1. Gastric Fluid and Bile Salts Tolerance

Strain survival rate to simulated digestive conditions, estimated by the mean cell viability, represents the tolerance of the different strains to gastric fluid (pepsin 1.33%; pH 2) and to bile salts (0.5%), respectively. The result is shown in Figure 3. The 2-factor ANOVA test showed a very high significant difference (*p* < 0.001) between the nine strains with respect to tolerance to the digestive conditions studied. The viability of the strains was higher in the bile salts with survival rates varying between 94.84% and 107.17% throughout the 240 min of the incubation. In contrast, the strain’s viability rate with respect to gastric fluid was slightly lower and ranged from 80.37% to 101.27% after the same incubation time. According to the results of Tukey’s tests, the strains’ viability-rate averages recorded after 240 min of exposure to gastric fluid and bile salts conditions are classified into homogeneous groups according to Tukey’s significant difference (*p* < 0.05). The two strains LbN1 and LbN14 were significantly the most tolerant to bile salts with viability rates of 107.17% and 105.17%, respectively. LbN14 was also the most tolerant strain to gastric fluid with a viability rate of 101.27%; on the other hand, LbN1 was less tolerant with a viability rate of 81.35%. There is no significant difference between the two strains LbN11 and LbN9 in the two tolerance tests. The strains LbN10, LbN11, LbN12, and LbN9 form a group of medium bile salt tolerance compared to the other strains. In contrast, regarding gastric fluid tolerance, the medium tolerance group is represented by LbN9, LbN11, LbN13, and LbN15 strains.

Strain tolerance to digestive enzymes, such as trypsin and pancreatic amylase, was used to evaluate their survival. The results obtained are presented in Figure 4. Tukey’s test (*p* < 0.05) classified the strains into four homogeneous groups with respect to enzyme tolerance. All the strains studied showed good viability greater than 90% after 5 h of exposure. There was no significant difference between the LbN1, LbN14, and LbN15 strains, which formed the best trypsin-tolerant group with viability rates of 96, 97.94, and 96.14%, respectively. These same strains also formed the best pancreatic amylase tolerant group with viability rates of 105.07, 99.74, and 99.47% for LbN1, LbN14, and LbN15, respectively. Therefore, these results suggest that the strains can survive in the human gut environment.

#### 3.2.2. Antioxidant Activity

The antioxidant activity of *L. plantarum* strains was determined by the DPPH radical scavenging test. The ANOVA results show a very highly significant difference (*p* < 0.001) between the nine strains with respect to the inhibition of DPPH radical (Figure 5). The Tukey post hoc test was used to classify the strains into homogeneous groups according to their antiradical activity. There was no significant difference between the recorded free radical inhibition percentages of the seven strains LbN15, LbN5, LbN11, LbN01, LbN9, LbN14, and LbN10; ranging from 62.8% to 92.15%. The weakest antiradical activities were recorded by LbN13 (54.96%) and LbN12 (54.35%), which were not significantly different.

#### 3.2.3. Antimicrobial Activity

The results of the disk diffusion test to determine the diameters of the zones of inhibition of the antimicrobial activity of the strains of *L. plantarum* are reported in Table 2. The results of the MANOVA test reveal a significant difference between the different strains with regard to the inhibition of the growth of the different pathogens tested.

The Tukey test was undertaken to better appreciate the comparison between the effectiveness of the strains studied. The minimal and maximal inhibition zone diameters recorded in this work are 4.9 and 17.47 mm, respectively. The nine strains showed a large inhibition zone diameter against the pathogenic bacteria tested, except against *Aspergillus niger* ATCC 106404 whose best zone of inhibition is 9 mm (LbN9). Most strains exhibited the same effect on *Bacillus subtilis* ATCC 6633, with inhibition zone diameters ranging from 13.10 to 14.67 mm, except those of LbN5 and LbN13, whose inhibition zones diameters were 12.07 mm and 11.58 mm, respectively. The *L. plantarum* strains were classified into three homogeneous groups with respect to the growth inhibition of *Staphylococcus aureus* ATCC 33862. The largest inhibition zone diameter (14.27 mm) was obtained with LbN14. The strains had an antibacterial effect on both gram-negative and -positive bacteria, but the effect of these strains on *Escherichia coli* ATCC 25922 was less than that exerted by the different strains on *Pseudomonas aeruginosa* ATCC 27853 with a maximum inhibition zone diameter of 17.33 mm obtained with LBN15. The largest inhibition zone diameter (17.47 mm) was recorded against *Candida albicans* ATCC10231 with the LbN14 strain.

## 4. Discussion

Raw milk (from cows, goats, buffalos, sheep, camels, and donkeys), as well as human breast milk, are sources of heterogeneous lactic flora, commonly used to isolate new LAB strains with important probiotic potential, which aroused the interest of several research teams around the world [1,25,26,27]. LABs have a GRAS (Generally Recognized As Safe) status and many probiotic properties, giving them a multitude of opportunities to be used in the food industry [28]. Therefore, the present study seeks to isolate new LAB strains from Algerian cow’s milk in order to select those with probiotic properties of interest.

There were nine selected isolates of gram-positive, catalase-negative, and heterofermentative. The 16s rDNA and MALDI-TOF MS showed that these isolates belong to lactobacilli genera, namely *Lacticaseibacillus rhamnosus* ”LbN1”, *Lactiplantibacillus plantarum* “LbN5, LbN9, LbN10, LBN11, LbN14, and LbN15” and *Lactiplantibacillus plantarum* subsp. *Argentoratensis* ”LbN12”. Moreover, there was a species diversity for some isolates that were related to several different species of the same genus and the same percentage. The isolates LbN5, LbN9, LbN10, LbN11, LbN13, LbN14, and LbN15 were 100% identified as *Lactiplantibacillus plantarum* with the same homology rate as that of the following species: *L. pentosus, L. herbarum, L. paraplantrum, L. fabifermentans*, and *L. argentoratensis*. The isolate LbN12 has shown 99.75% similarity to *Lacticaseibacillus* plantarum, and *L. pentosus*, *L. herbarum*, *L. paraplantrum*, *L. fabifermentans*, and *L. argentoratensis*, respectively; while the LnB1 isolate similarity to *Lacticaseibacillus rhamnosus,* L. casei, L. paracasei, and *L. paracasei subsp tolerans* was about 99.75%. The LbN1 strain belongs to *Lacticaseibacillus rhamnosus* according to the recent *Lactobacillus* taxonomy changes [29]. This classification creates a new opportunity for scientific discovery and reveals that the genus *Lactobacillus* includes 261 species (as of March 2020) that are extremely diverse at the phenotypic, ecological, and genotypic levels. It also proposed the reclassifying of the *Lactobacillus* genus into 25 genera in the *Lactobacillus* genus, signed under group name *Lactobacillus delbrueckii, Paralactobacillus* and 23 new genera represented with the names *Holzapfelia, Amylolactobacillus, Bombilactobacillus, Companiolactobacillus, Lapidilactobacillus, Agrilactobavillus, Schleiferilactobacillus, Loigolactobacillus, Lacticaseibacillus, Latilactobacillus, Dellaglioa, Liquorilactobacillus, Ligilactobacillus, Lactiplantibacillus, Furfurilactobacillus, Paucilactobacillus, Limosilactobacillus, Fructilactobacillus, Acetilactobacillus, Apilactobacillus, Levilactobacillus, Secundilactobacillus* and *Lentilactobacillus* [29].

The proteomic results obtained by the MALDI-TOF spectrometry technique were coherent with rRNA 16S identification. Both methods lead to the same result with respect to *Lactobacillus plantarum* versus *Lactiplantibacillus plantarum*. This consistency between the two methods cited was also observed by many other authors who have underlined the MALDI-TOF performance confirming the results of identification of Lactobacillus strains by genotyping methods [1,13].

To evaluate the probiotic potential of lactobacilli strains, some criteria were investigated in this work. All strains exhibited good tolerance to the in vitro tests of simulated host gastrointestinal conditions and good resistance against digestive enzymes. Several studies have recorded good viability of LABs, especially L. plantarum under different conditions of the gastrointestinal tract. Indeed, Iorizzo et al. [8] showed a high survival rate for 10 strains of L. plantarum in the presence of 0.3% of bile salts in an acidic medium (pH 2.5). Fidanza et al. [30] reported that the tolerance of L. plantarum species to extreme conditions of the gastrointestinal tract is explained by the use of several mechanisms, such as the rapid recycling of damaged proteins and the expression of phosphofructokinase (pfk) and pyruvate kinase (pyk) for carbohydrate metabolism. These mechanisms play an important role in maintaining intracellular pH for the cellular capacity for homeostasis and promote resistance to stress from environmental factors. Moreover, strains with probiotic potential must exhibit bile salt tolerance, such as the expression of bile salt hydrolase (BSH), which catalyzes the deconjugation of glycine and taurine residues from cholesterol [31] and has been definitively linked to the tolerance of a number of Lactobacilli species exposed to bile salts [30]. Additionally, the amount of probiotics surviving in transit through the human gut depends on the strain, ingested dose, host-related stressors (acidity, bile salts, etc.), and food carrier [32]. Additionally, Chen et al. [33] revealed for the first time that five new bile salt tolerance-related genes involved in the phosphotransferase system, the two-component system, the carbohydrate metabolic pathway, and malate dehydrogenase can be used as a biomarker to screen for bile-salt-resistant lactobacilli [33].

All the strains showed an important antioxidant activity with the highest DPPH scavenging activity (92.15%) being obtained with the LbN09 strain. These results are in agreement with those of Düz et al. [34] who reported that the highest antioxidant activity exhibited by *L. plantarum* IH14L was 90.34%, while the lowest percentage (58.38) was recorded with the GH8L strain. In the work of Won et al. [35], the highest DPPH radical scavenging activity was recorded for *Lactiplantibacillus plantarum* (60%) and *Lacticaseibacillus paracasei* (75.80%). The antioxidant activity can be explained by the production of cell surface compounds such as strain-produced extracellular polysaccharides and lipoteichoic acid and the existence of enzymes, such as NADH-oxidase, SOD, NADH peroxide and non-heme manganese catalases in LAB bacteria [34,36].

The antimicrobial or antagonistic activity is an important criterion for the selection of probiotics. The production of antimicrobial compounds like bacteriocins, competitive exclusion of pathogens, strengthening the intestinal barrier against pathogens, and boosting the host’s immune system to fight off pathogens are all crucial properties of significance [37]. All bacteria exhibit an antibacterial activity strain with inhibition zone diameters ranging from 4.9 mm against *Aspergillus niger* ATCC 106404 to 17.47 mm against *Candida albicans* ATCC 1023. These results are similar to those of Jiang et al. [4] who showed that *L. plantarum* WLPL04 had broad-spectrum activity against gram-positive strains including *Staphylococcus aureus* CMCC26003 and gram-negative strains such as *Pseudomonas aeruginosa* MCC10104. Neutralization of the supernatant permits the elimination of the organic acids effect and indicates the possible existence of other inhibitory agents produced by the lactobacilli tested, which may be hydrogen peroxide, diacetyl, or bacteriocins. Chaalel et al. [22] reported that a *Lactobacillus plantarum* LbM2a strain, isolated from the feces of one-week-old newborns, produced a plantaricin substance that allowed the inhibition of 18 of the 22 indicator strains, including *Staphylococcus aureus, Escherichia coli*, and *Bacillus cereus.*

## 5. Conclusions

A total of nine strains of lactobacilli were isolated from raw cow’s milk collected in North-West of Algeria. The combination of genotypic and biomolecular analyses revealed the presence of three different species of lactobacilli bacteria, namely *Lacticaseibacillus rhamnosus* (LbN1)*, Lactiplantibacillus plantarum* (LbN5, LbN9, LbN10, LBN11, LbN14, LbN15) and *Lactiplantibacillus plantarum* subsp. *Argentoratensis* (LbN12). The convergence of these two results confirms the reliability of the MALDI-TOF MS as a method of identification. Isolated strains exhibit an appreciable resistance to the host simulated gastric acidity and intestinal conditions (i.e., bile salts and digestive enzymes), and they have relevant antioxidant and antimicrobial activities. The results of this experiment are conclusive regarding the identity of the strains and the in vitro tests of candidacy for probiotic status undergone by the isolated strains. Such properties are in favor of further investigations to strengthen the probiotic status of these strains in order to be used in the food industry.

## Figures and Tables

**Figure 1 microorganisms-11-02091-f001:**
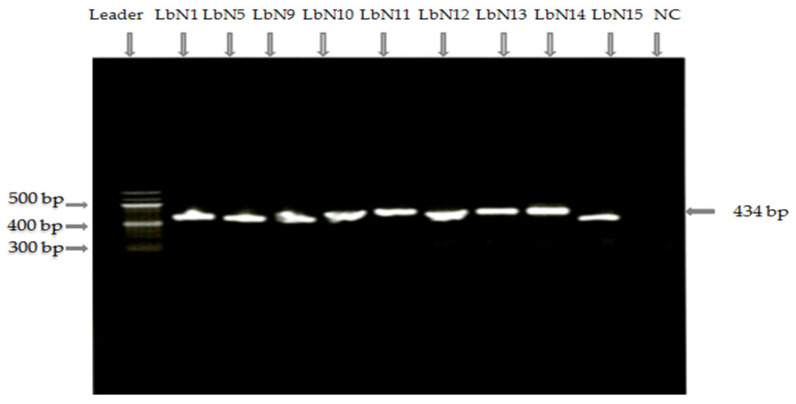
A 1% Agarose gel electrophoresis made with TBE 1X solution (TBE 1X = 10 mL of distilled water make up to 1L de TBE 10X) of PCR rRNA 16S using the universal primers P3:Bact968-GC-f and P4: Bact1401-r, related to hypervariable regions V6-V8 region with fragment length 434 bp. Leader; NC: Negative control; LbN1, LbN5, LbN9, LbN10, LbN11, LbN12, LbN13, LbN14, and LbN15: Isolates.

**Figure 2 microorganisms-11-02091-f002:**
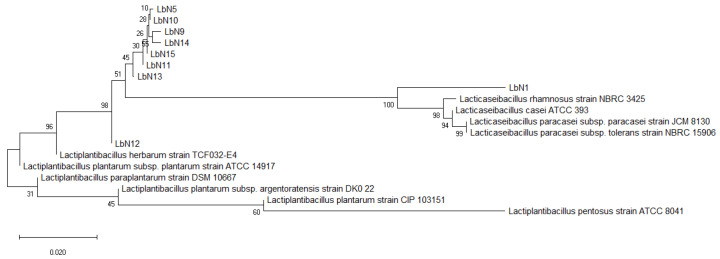
Dendrogram based on the 16S rRNA gene sequences, showing the relationships within the lactobacillus isolates and their positions with representatives of some other related taxa from the NCBI database (*Lacticaseibacillus rhamnosus* NBRC 3425, *Lacticaseibacillus casei* ATCC 393, *Lacticaseibacillus paracasei* subsp. *paracasei* CM 8130, *Lacticaseibacillus paracasei* subsp *tolerans* NBRC 15906, *Lactiplantibacillus herbarum* TCFE 032-E4, *Lactiplantibacillus parapalantarum* DSM 10667, *Lactiplantibacillus plantarum* subsp. *Argentoratensis* DK 022, and *Lactiplantibacillus pentosus* strain ATCC 8041). The sequences were aligned using the neighbor-joining method with Jukes-Cantor correction. Bootstrap values are shown on the nodes.

**Figure 3 microorganisms-11-02091-f003:**
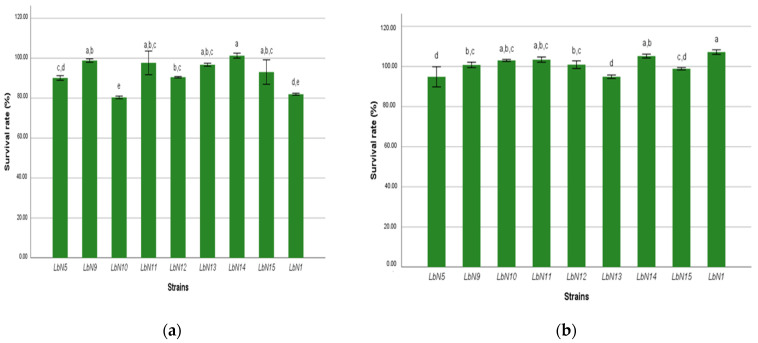
Strain viability rate to gastric fluid and bile salts at 240 min: (**a**) gastric fluid (pepsin 1.33%); (**b**) bile salts 0.5%. (ANOVA, *p* < 0.001). a,b,c,d,e: homogeneous groups.

**Figure 4 microorganisms-11-02091-f004:**
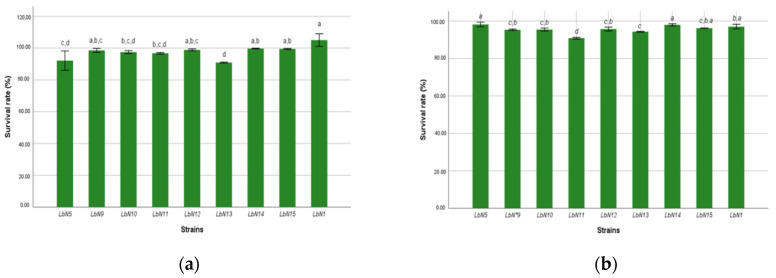
Viability rate of strains to intestinal fluid: (**a**) pancreatic amylase; (**b**) Trypsin. (ANOVA, *p* < 0.001). a,b,c,d: homogeneous groups.

**Figure 5 microorganisms-11-02091-f005:**
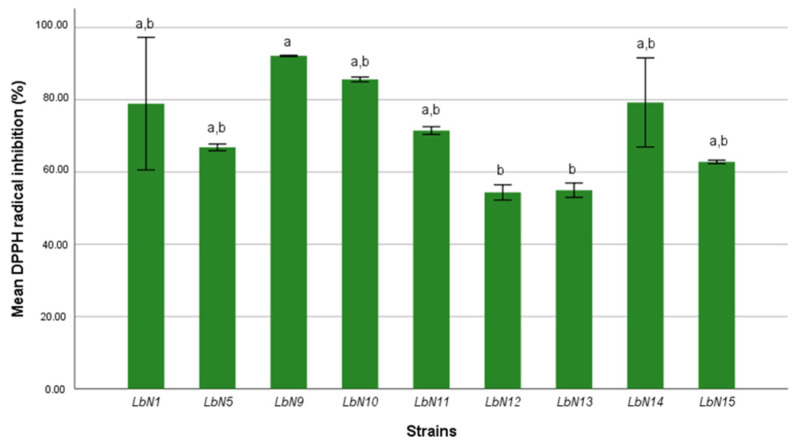
Antioxidant activity of strains of *L. plantarum* estimated by DPPH radical scavenging activity (ANOVA, *p* < 0.001). a,b: homogeneous groups.

**Table 1 microorganisms-11-02091-t001:** Similarity scores (%) of identification results by 16S rRNA of lactic acid bacteria isolates.

Isolates Code	Species Identification	Similarity Scores (%)
LbN1	*Lacticaseibacillus rhamnosus*	99.75
LbN5	*Lacticaseibacillus plantarum*	100
LbN9	*Lacticaseibacillus plantarum*	100
LbN10	*Lacticaseibacillus plantarum*	100
LbN11	*Lacticaseibacillus plantarum*	100
LbN12	*Lacticaseibacillus plantarum*	99.75
LbN13	*Lacticaseibacillus plantarum*	100
LbN14	*Lacticaseibacillus plantarum*	100
LbN15	*Lacticaseibacillus plantarum*	100

**Table 2 microorganisms-11-02091-t002:** Antimicrobial activities of isolated LAB strains against six food-borne pathogenic microorganisms. a,b,c,d,e: homogeneous groups.

Isolated LAB Strains	Pathogens/Inhibition Zone (mm)					
	*B. subtilis* ATCC 6633	*P. aeruginosa* ATCC 27853	*E. coli* ATCC 25922	*S.aureus* ATCC 33862	*C. albicans* ATCC 10231	*A. niger* ATCC 106404
LbN1	13.10 ± 0.36 ^a,b,c^	14.20 ± 0.72 ^b^	12.50 ± 0.5 ^a,b^	12.03 ± 0.15 ^b,c^	14.67 ± 0.58 ^c,d^	6.83 ± 0.76 ^b,c,d,e^
LbN5	12.07± 0.11 ^b,c^	14.17± 0.28 ^b^	11.97 ± 0.95 ^a,b^	13.50 ± 0.50 ^a,b^	17.00 ± 1.00 ^a,b^	8.73 ± 0.46 ^a,b^
LbN9	13.83 ± 0.28 ^a,b^	14.40 ± 1.21 ^b^	13.80 ± 0.38 ^a^	13.40 ± 0.52 ^a,b^	14.17 ± 0.28 ^d^	9.00 ± 0.50 ^a^
LbN10	14.57 ± 0.40 ^a^	15.03 ±0.55 ^a,b^	13.10 ± 0.56 ^a,b^	12.23 ± 1.15 ^b^	15.77 ± 0.49 ^b,c^	8.07 ± 1.10 ^a,b,c,d^
LbN11	14.67 ± 0.57 ^a^	14.80 ± 0.26 ^b^	12.10 ± 0.26 ^a,b^	12.97 ± 0.45 ^a,b^	15.60 ± 0.60 ^b,c,d^	4.90 ± 0.36 ^e^
LbN12	14.50 ± 0.50 ^a^	14.33 ± 1.15 ^b^	11.37 ± 0.78 ^b^	12.63 ± 0.32 ^a,b^	14.60 ± 0.52 ^c,d^	8.33 ± 0.58 ^a,b,c^
LbN13	11.58 ± 0.95 ^c^	16.17 ± 0.76 ^a,b^	12.40 ± 0.52 ^a,b^	10.27 ± 0.64 ^c^	16.83 ± 0.28 ^a,b^	7.13 ± 0.32 ^a,b,c,d^
LbN14	14.49 ± 1.32 ^a^	15.77 ± 0.93 ^a,b^	12.80 ± 0.72 ^a,b^	14.27 ± 1.10 ^a^	17.47 ± 0.45 ^a^	6.23 ± 1.12 ^d,e^
LbN15	14.47 ± 0.55 ^a^	17.33 ± 1.15 ^a^	12.83 ± 1.60 ^a,b^	13.80 ± 0.20 ^a,b^	14.87 ± 0.41 ^c,d^	6.43 ± 0.51 ^c,d,e^
*p* value * (MANOVA test)	0.000	0.003	0.04	0.000	0.000	0.000

* Each value represents the mean ± S.D. Statistical analysis was performed using the MANOVA test followed by Tukey’s test.

## Data Availability

Not applicable.

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
