# Peer review of "Probiotic Properties of Lactic Acid Bacteria Newly Isolated from Algerian Raw Cow’s Milk"

_microorganisms, 2023, doi:10.3390/microorganisms11082091_

Round 1

Reviewer 1 Report

This manuscript described the isolation and identification of the Lactobacilli strains from algerien cows milk by 16S rRNA gene sequence analysis and MALDI-TOF mass spectrometry method, and evaluation their mainly probiotic properties by in vitro determination of the survival to pass though the simulated gastrointestinal conditions, antimicrobial and antioxidant activities. This manuscript shown potential application of those isolutes in food industries. But the following issues should be concerns:

1.     Line 166~176, Antimicrobial activities ”the plates were incubated at 37overnight” this condition is only suitable for bacterial growth, but did not for Aspergillus niger, Candida albicans. Please check it.

2.     Line 124~128the accession Number of the nine isolates should be moved to the results in line 201~202.

3.     In the result section. The title 2.2.1 were longer, please simplify it.

4.     In Fig3, the controls were lacked in which the cell pellets in salt solution as controls to incubated 240min.

5.     In Fig4, the controls were lacked in which the cell pellets were incubated with inactived enzymes including in pancreatic amlylase, trypsin as controls.

6.     Fig 3, Fig4, Fig5 and Table 2 should be drawn according to the instruction of this Journal.

7.     Line 305~309, please supplying the pictures of the inhibition zones of the isolate LbN9 or LbN14 against Aspergillus niger, Candida albicans.

8.     The results in Abstract was not accorded with the result in result section 3.2.3, “the nine strains shown antibacterial effects on all the pathogens tested expected for As. niger ATCC 106404……”

Reviewer 2 Report

Dear Editor and Authors,

I send you my review about the article entitled Probiotic Properties of Lactic Acid Bacteria Newly Isolated from Algerian Raw Cow’s Milk”.

The aim of the paper, as reported in the scope was to screen new lactic acid bacteria from Algerian cow's milk to assess their probiotic properties.

In my opinion, although the Article is well written, in a good English language and it is well structured, it show, also, some lacks that I reported below.

In the introduction should be better explained the originality of this Article. To improve this aspect, I suggest to the Authors, to stress the relevance of the new strains of probiotics. Moreover it should stressed the difference among this research and the previous ones reported in literature.

The paragraph materials and methods result complete, but the number of milk samples collected should be reported.

Moreover, if it possible, it should be reported some data about the herd or the herds were the samples were collected.

Furthermore, to facilitate the understanding by the readers, the paragraph “2.3” should be merged with the previous paragraph “2.2.”.

Again, also the sub-paragraph 2.5 should be merged in a single paragraph.

The results is well presented and they are well discussed, also in comparison to the data reported in the literature.

Finally, the conclusions appear to be too synthetic respect the data showed and to the aim of the research, in my opinion, it should best stress the relevance of the findings of this research.

Best regards

Author Response

Point1 :  In the introduction should be better explained the originality of this Article. To improve this aspect, I suggest to the Authors, to stress the relevance of the new strains of probiotics. Moreover, it should stressed the difference among this research and the previous ones reported in literature.

Response 1: The introduction was slightly modified. Three new sentences were added  and highlighted in red in the manuscript.

Point 2 : The paragraph materials and methods result complete, but the number of milk samples collected should be reported.Moreover, if it possible, it should be reported some data about the herd or the herds were the samples were collected.

Response 2 :The number of milksamples and information about herd (cowfeed) werereported in paragraph 2.1, and highlighted in red in manuscript.

Point 3 : Furthermore, to facilitate the understanding by the readers, the paragraph “2.3” should be merged with the previous paragraph “2.2.”.

Response 3 : The paragraphs 2.2 and 2.3 were merged

Point 4: Again, also the sub-paragraph 2.5 should be merged in a single paragraph.

Response 4: It was done and highlighted in red in the manuscript.

Point 5:  Finally, the conclusions appear to be too synthetic respect the data showed and to the aim of the research, in my opinion, it should best stress the relevance of the findings of this research.

Response 5: The relevant findings of this experiment have been included in the conclusion.